# Multiple sclerosis and COVID-19: A retrospective study in Iran

Behnaz Sedighi[1], Aliakbar Haghdoost[2], Parya Jangipour Afshar[3], Zohre Abna[4], Shamimeh Bahmani[5], Simin Jafari[1,5]*

1 Neurology Research Center, Kerman University of Medical Sciences, Kerman, Iran, 2 Institute for Futures Studies in Health, Modeling in Health Research Center, Kerman University of Medical Sciences, Kerman, Iran, 3 Faculty of Public Health, Department of Biostatistics and Epidemiology, Kerman University of Medical Sciences, Kerman, Iran, 4 Tehran University of Medical Sciences, Tehran, Iran, 5 Faculty of Medicine, Kerman University of Medical Sciences, Kerman, Iran

* jafarisimin93@gmail.com

**Data Availability Statement:** All relevant data are within the paper and its Supporting Information files.

**Funding:** The author received no specific funding for this work.

## Abstract

### Objectives

Previous studies suggested a higher rate of COVID-19 infection in patients with multiple sclerosis than in the general population, and limited studies addressed the impact of COVID-19 and its vaccination in patients with multiple sclerosis in Iran. We decided to investigate the factors associated with COVID-19 infection, the effects and side effects of the COVID-19 vaccination in patients with multiple sclerosis (MS).

### Methods

We used the data of the patients with multiple sclerosis registered in a referral clinic in Kerman, one of the large cities in Iran (a population of 537,000 inhabitants), to explore the association between demographic variables, the history of COVID-19 vaccination, and the clinical outcomes.

### Results

Of the 367 participants in this study, 88.3% received the COVID-19 vaccine, 35.4% were confirmed COVID-19 cases, and the incidence of COVID-19 was much higher before vaccination (24.5% before vaccination versus 10.1% after vaccination). The multivariable logistic regression model showed that male gender (OR = 2.64, 95% confidence interval: 1.21, 5.74) and current employment (OR = 3.04, 95% confidence interval: 1.59, 5.80) were associated with an increased risk of COVID-19. The only factor associated with the adverse effects of COVID-19 vaccination was the type of vaccine (AstraZeneca).

### Conclusion

Our findings showed that the vaccination protected MS cases considerably against COVID-19. In addition, the side effects of the vaccines were not noticeably high in these cases as well. Among all COVID-19 vaccines, AstraZeneca had the most common side effects, so people must be aware of them before vaccination. The male gender and employment were

**Competing interests:** The authors have declared that no competing interests exist.

the most important variables in the prevalence of COVID-19 in patients with multiple sclerosis in our study.

## Introduction

Severe acute respiratory syndrome coronavirus 2 (SARS-CoV-2) is responsible for COVID-19, which was first seen in Wuhan, China, in 2019 [1]. This disease is very contagious and manifests with fever, cough, myalgia, and dyspnea. The virus spread all over the world, resulting in a pandemic that killed 5.4 million people and infected more than 304 million people until January 9, 2022 [2]. COVID-19 posed a great challenge to all healthcare systems around the world, with the greatest impact on the health organizations of each country [3].

Multiple sclerosis (MS), one of the most disabling CNS diseases, is an autoimmune inflammatory demyelinating pathology that affects approximately 2.8 million people worldwide [4,5]. Many patients with multiple sclerosis have a suppressed immune system due to disease-modifying therapy (DMT), so they are more at risk for infectious diseases than the general population [6].

The COVID-19 outbreak has caused many issues for patients with multiple sclerosis in various aspects of their lives, including fragile mental health, poor treatment follow-up, and increased annual hospitalization [7,8]. The suppressive immune system due to DMT has also raised concerns about the facilitation of COVID-19 infection in patients with MS [9–11].

Fortunately, initiating vaccination against COVID-19 has controlled the spread of the disease and improved the medical status of patients with MS [12,13]. Previous studies showed that COVID-19 has a higher rate of infection and hospitalization in patients with MS than in the general population, while early vaccination may be effective in reducing these rates [14]. Numerous studies were conducted on the patients with MS in Iran during the COVID-19 pandemic [7,8,15]. However, limited studies investigated the factors affecting COVID-19 infection, the effectiveness of vaccination, and the complications of vaccination in patients with MS in Iran. Most of these studies were conducted at the beginning of the pandemic [14,16–18]. The current study investigated the factors associated with COVID-19 infection, the impact of vaccination, as well as factors affecting the side effects of vaccination in patients with MS in Kerman Province, Iran.

## Materials and methods

### Study design and participants

This retrospective study aimed to investigate the factors associated with the COVID-19 and the effectiveness and side effects of vaccination in patients with MS in a referral clinic in Kerman, one of the large cities in Iran, from March to December 2021.

We used the data of patients registered in the referral hospital in Kerman province. Out of 548 registered cases, we recruited 367 (66.9%) and excluded those who disagreed or had a major non-communicable disease, such as diabetes, hypertension, cardiovascular, cerebrovascular, and respiratory diseases. The final diagnosis of MS in this registry was confirmed by neurologists who are affiliated with the Shafa hospital and the Kerman University of Medical Sciences, based on the revised McDonald criteria (2017) [19].

### Outcomes measurement

Data were collected using three questionnaires: 1. demographic information (age, gender, BMI, education level, employment status, number of children, place of residence (urban or

rural), no common comorbidities (complications accompanying MS other than diabetes, hypertension, cardiovascular, cerebrovascular, and respiratory diseases), tobacco, alcohol, and substance abuse (opioids, heroin, methamphetamine, or other stimulants)); 2: COVID-19-related information (including vaccine injection, type, and side effects; confirmed COVID-19 that was based on physician judgment, which used lab tests including PCR and clinical signs and symptoms since the beginning of the pandemic; infection with COVID-19 before or after vaccination; and the time (days) to infection after vaccination); 3: Multiple sclerosis-related information (duration of illness and the Expanded Disability Status Scale).

The expanded disability status scale (EDSS) of MS patients was designed by John Kurtzke. It assesses the functioning of systems such as pyramidal, cerebellar, brainstem, sensory, bowel, and bladder, visual and cerebellar regions [20], The score varies from 0 (a normal neurological state) to 10 (MS-induced death): 0–2.5 refers to people with a mild degree of disability, 3–5 refers to people with a moderate degree of disability, and 5.5–10 refers to people with a severe degree of disability.

## Statistical analysis

SPSS 26 was used to analyze the data. Descriptive statistics were used to describe qualitative (number and percentage) and quantitative variables (mean and standard deviation). A multi-variable logistic regression model was used to determine the factors related to COVID-19 and its side effects in patients with MS. A significant level of 0.05 was considered.

## Ethics statement

This study was approved by the Ethics Committee of Kerman University of Medical Sciences (IR.KMU.REC.1400.566). Written informed consent was obtained from all the participants.

## Results

The study included 367 participants, of whom 305 (83.1%) were female and 62 (16.9%) were male. Participants were between 17 and 67 years old, with a mean age of 37.99±9.80. The mean duration of multiple sclerosis was 8.26 ± 5.60 years, and the mean EDSS score was 1.18±1.60 out of 10, which was categorized as a mild category. Most of the patients had 12–16 years of academic education (80.4%), were unemployed (67.8%), lived in urban areas (93.5%), did not use tobacco or other substances (89.4%), received a vaccine against COVID-19 (88.3%), did not acquire COVID-19 (64.6%), and had mild disease activity (84.2%). Tables 1 and 2 present other demographic characteristics.

The multivariable logistic regression model showed that gender (OR = 2.64, 95% confidence interval: 1.21, 5.74), and employment status (OR = 3.04, 95% confidence interval: 1.59, 5.80), were all associated with an increased risk of COVID-19 infection in the patients with MS. (Table 3).

According to a multivariable logistic regression model, only the type of vaccine (AstraZeneca) (OR = 23.578, 95% confidence interval: 3.041,182.80) was a risk factor for adverse effects of COVID-19 injection in the patients with MS (Table 4).

We compared the frequency of post-vaccination side effects among patients based on the types of vaccines. Our results showed that the patients with multiple sclerosis who received AstraZeneca and Sinopharm had significant adverse effects (Table 5). Also, our results showed that the patients had side effects such as injection site pain, tenderness, redness, fever, headache, fatigue, nausea, diarrhea, and muscle pain after vaccination. The most common adverse effect was pain at the injection site (31.3%) (Table 6).

**Table 1. Demographic characteristics (N = 367).**

| Variables | group | Number | Percent |
|---|---|---|---|
| **Gender** | Female | 305 | 83.1 |
| | Male | 62 | 16.9 |
| **Education(years)** | <12 | 45 | 12.3 |
| | 12–16 | 295 | 80.4 |
| | >16 | 27 | 7.4 |
| **Number of children** | none | 90 | 24.5 |
| | ≤2 | 211 | 57.5 |
| | >2 | 66 | 18.0 |
| **Job** | Unemployed | 249 | 67.8 |
| | Self-employed | 49 | 13.4 |
| | Employed | 69 | 18.8 |
| **Location** | Urban | 343 | 93.5 |
| | Rural | 24 | 6.5 |
| **comorbidities** | No | 309 | 84.2 |
| | Yes | 58 | 15.8 |
| **Addiction** | No | 328 | 89.4 |
| | Yes | 38 | 10.4 |
| **Vaccination** | No | 43 | 11.7 |
| | Yes | 324 | 88.3 |
| **Adverse effects after vaccination** | No | 175 | 47.7 |
| | Yes | 143 | 39.0 |
| **Getting COVID-19** | No | 237 | 64.6 |
| | Yes | 128 | 34.9 |
| **Time of catching COVID-19** | Before injection | 90 | 24.5 |
| | After injection | 37 | 10.1 |
| **EDSS** | (0–2.5) Mild | 309 | 84.2 |
| | Moderate (3–5) | 44 | 12.0 |
| | Sever (5.5–10) | 13 | 3.5 |
| **DMT** | Injectable | 136 | 37.05 |
| | Anti CD20 | 63 | 17.17 |
| | S1P receptor modulator (Fingolimod) | 43 | 11.72 |
| | Natalizumab | 18 | 4.91 |
| | Oral (Dimethyl fumarate, Teriflunomide) | 71 | 19.35 |
| | None | 36 | 9.80 |

The sum of subgroups may be less than total because of missing data.

DMT: Disease-modifying therapy, S1P: Sphingosine 1-phosphate, EDSS = Expanded Disability Status Scale.

Furthermore, we compared the prevalence of COVID-19 in patients with MS based on the type of DMT, and our results showed no significant difference in the prevalence of COVID-19 based on the kind of DMT taken by the patients. (Table 7).

## Discussion

We investigated the factors associated with COVID-19 infection, the impacts and side effects of vaccination in patients with multiple sclerosis in Kerman Province, Iran.

We found that the male gender and employment status of the patients with MS played a significant role in evaluating factors associated with suspected or confirmed COVID-19, and the

**Table 2. Mean and standard deviation of variables.**

| Variables | Mean ±SD | Min | Max |
|---|---|---|---|
| Age (year) | 37.99±9.80 | 17 | 67 |
| The duration of MS (year) | 8.26±5.60 | 1 | 30 |
| BMI (kg/m2) | 24.69±4.08 | 15 | 39.84 |
| Time between injection and catching COVID-19 (day) | 39.56±25.22 | 2 | 120 |
| EDSS | 1.18±1.60 | 0 | 7 |

BMI = body mass index, EDSS = Expanded Disability Status Scale.

**Table 3. Logistic regression analysis of factors associated with COVID-19.**

| Subgroup | Odds ratio (95% CI) | P-value |
|---|---|---|
| **Gender** | | |
| Female | Reference | |
| Male | 2.64 (1.21,5.74) | 0.01* |
| **Education(years)** | | |
| <12 | Reference | |
| 12–16 | 0.82 (0.38,1.76) | 0.61 |
| >16 | 0.74 (0.23,2.34) | 0.61 |
| **Job** | | |
| Unemployed | Reference | |
| Self-employed | 1.113 (0.50,2.47) | 0.79 |
| Employed | 3.041 (1.59,5.80) | 0.00* |
| **Location** | | |
| Urban | Reference | |
| Rural | 0.47 (0.18,1.23) | 0.12 |
| **Vaccination** | | |
| No | Reference | |
| Yes | 0.61 (0.30,1.24) | 0.17 |
| **Addiction** | | |
| No | Reference | |
| Yes | 1.09 (0.48,2.50) | 0.82 |
| **comorbidities** | | |
| No | Reference | |
| Yes | 1.26 (0.66,2.39) | 0.46 |
| **EDSS** | | |
| Mild (0–2.5) | Reference | |
| Moderate (3–5) | 1.05 (0.47,2.31) | 0.90 |
| Sever (5.5–10) | 2.08 (0.52,8.21) | 0.29 |
| **Age** | 0.98 (0.95,1.01) | 0.24 |
| **BMI (kg/m$^2$)** | 1.02 (0.96,1.08) | 0.35 |
| **The duration of MS (year)** | 1.02 (0.97,1.07) | 0.31 |

The sum of subgroups may be less than total because of missing data.

BMI = body mass index; CI = confidence interval; EDSS = Expanded Disability Status Scale.

*Is significant at P<0.05.

**Table 4. Logistic regression analysis of factors associated with adverse effect of vaccination.**

| Subgroup | Odds ratio (95% CI) | P-value |
|---|---|---|
| **Gender** | | |
| Female | Reference | |
| Male | 0.75 (0.35,1.59) | 0.46 |
| **Education (years)** | | |
| <12 | Reference | |
| 12–16 | 0.84 (0.39,1.81) | 0.66 |
| >16 | 0.93 (0.29,2.97) | 0.91 |
| **Job** | | |
| Unemployed | Reference | |
| Self-employed | 0.92 (0.41,2.08) | 0.85 |
| Employed | 0.67 (0.34,1.31) | 0.25 |
| **Location** | | |
| Urban | Reference | |
| Rural | 0.49 (0.15,1.51) | 0.21 |
| **EDSS** | | |
| Mild (0–2.5) | Reference | |
| Moderate (3–5) | 0.88 (0.40, 1.91) | 0.75 |
| Sever (5.5–10) | 0.57 (0.13,2.52) | 0.46 |
| **EDSS** | | |
| No | Reference | |
| Yes | 0.94 (0.48,1.82) | 0.86 |
| **Addiction** | | |
| No | Reference | |
| Yes | 0.66 (0.28,1.53) | 0.33 |
| **Type of vaccine** | | |
| Sinopharm | Reference | |
| Barekat | 0.56 (0.13,2.27) | 0.33 |
| AstraZeneca | 23.57 (3.04,182.80) | 0.00* |
| Sputnik | 0.82 (0.14,4.80) | 0.83 |
| **Age** | 0.98 (0.95,1.01) | 0.38 |
| BMI (kg/m$^2$) | 1.05 (0.99,1.12) | 0.06 |
| **Duration of disease** | 1.00 (0.96,1.05) | 0.74 |

The sum of subgroups may be less than total because of missing data.

BMI = body mass index; CI = confidence interval; EDSS = Expanded Disability Status Scale.

*Is significant at P<0.05.

**Table 5. Type of vaccine and adverse effect of vaccine in each group.**

| Vaccine | Adverse effect | Frequency | Percent | p-value |
|---|---|---|---|---|
| **AstraZeneca (n = 20)** | No | 1 | 5.0 | <0.00* |
| | Yes | 19 | 95.0 | |
| **Barekat (n = 11)** | No | 8 | 72.7 | 0.13 |
| | Yes | 3 | 27.3 | |
| **Sinopharm (n = 281)** | No | 162 | 56.8 | 0.01* |
| | Yes | 119 | 41.8 | |
| **Sputnik (n = 6)** | No | 4 | 66.7 | 0.41 |
| | Yes | 2 | 33.3 | |

*Is significant at P<0.05.

**Table 6. Frequency and percent of the adverse effect of vaccines.**

| Adverse effect | Frequency | percent |
|---|---|---|
| Injection site pain | 115 | 31.3 |
| Tenderness | 17 | 4.6 |
| Redness | 6 | 1.6 |
| Fever | 41 | 11.2 |
| Headache | 54 | 14.7 |
| Fatigue | 46 | 12.5 |
| Nausea | 36 | 9.8 |
| Diarrhea | 20 | 5.4 |
| Muscle pain | 36 | 9.8 |

The number of subgroups may be more than total (143) because some people may have more than one side effect.

type of vaccine (AstraZeneca) was the only factor associated with the adverse effects of COVID-19 vaccination.

We revealed that the prevalence of COVID-19 in the patients with MS was approximately 35%, but Naghavi et al. indicated that the suspected rate of COVID-19 in the patients with MS was 20.4%, with 11.7% having PCR confirmation [21], and Zabalza et al. reported that COVID-19 was prevalent in 6.3 percent of the patients with MS. The difference between these studies and our study could be due to genetic and racial differences, disease-modifying therapies, and poor compliance with COVID-19 protection protocols [22–26].

Another finding of this study indicated that 70 percent of the COVID-19 infection occurred before vaccination, supporting the fact that vaccination against COVID-19 can reduce the infection rate of the coronavirus. Large-scale vaccination could be the only way to prevent COVID-19 [12,27,28].

We discovered, like Moss and Jehi et al., that the male gender and employment could increase the prevalence of COVID-19 in MS patients [6,29]. But Naghavi et al. reported no association between gender and COVID-19. They found the presence of comorbidity, EDSS scores, DMT, hospitalization rate due to COVID-19 infection could be associated with gender [21]. Rostami et al. reported that autoimmune diseases, DMT, age, gender, and high EDSS increased COVID-19 incidence and hospitalization in patients with multiple sclerosis [24]. The high incidence of COVID-19 in men and employed people could be due to strict self-protection, self-isolation, and social distancing in women more than men, and also high exposure to the coronavirus in employed people.

In this study, we did not find any significant difference in contracting COVID-19 between users and non-users of DMT. And also, no association was found between COVID-19

**Table 7. Type of DMT and getting COVID-19 in each group.**

| DMT | Covid-19 | | P-value |
|---|---|---|---|
| | Yes | No | |
| Injectable | 49 | 86 | 0.14 |
| Anti CD20 | 23 | 40 | |
| S1P receptor modulator (Fingolimod) | 20 | 22 | |
| Natalizumab | 2 | 16 | |
| Oral (Dimethyl fumarate, Teriflunomide) | 23 | 48 | |
| None | 11 | 25 | |

prevalence and the type of DMT, even in patients receiving anti CD20 therapies. These findings are in line with Alonso and Parrotta et al. [30,31], which found no association between DMT and the risk of COVID-19. It could be due to the small sample size in this study, the younger participants, or the lower EDSS. Recent large studies have shown that patients with MS treated with anti-CD20 DMTs (rituximab or ocrelizumab) were at higher risk of developing severe COVID-19 than other DMTs [32,33]; however, strict adherence to health protocols and self-isolation might be an important reason for the low prevalence and severity of COVID-19 infection among our patients.

Studies suggest that the COVID-19 pandemic has affected patients with multiple sclerosis, so the development of multiple vaccines against COVID-19 could be promising for the control and eradication of the disease. According to the current study, 88% of the patients received at least one dose of the vaccine, indicating their willingness to receive COVID-19 vaccines; Also, in the patients on the anti B cell therapies, vaccination was done one month before taking the next dose of the drug or three months after taking the drug (as a result of the previous studies on the efficacy of the vaccines in the patients treated with these drugs). Previous studies found that patients with multiple sclerosis were less reluctant to receive the COVID-19 vaccines after healthcare providers prepared adequate information for them [27].

Furthermore, we showed that 39 percent of the patients had side effects such as injection site pain, tenderness, redness, fever, headache, fatigue, nausea, diarrhea, and muscle pain after vaccination. Most of them received Sinopharm vaccines, and a few of them received Barekat, AstraZeneca, and Sputnik vaccines. We found a significant relationship between the type of vaccine and side effects after vaccination; all vaccines had side effects, but AstraZeneca had the most common side effects. Zare et al. supported this result [34], in contrast Babaee and Omeish et al. rejected it [35,36]. Previous studies reviewed the effect of other vaccines on patients with MS and found that those on immunosuppressive drugs experienced serious side effects and a relapse after vaccination; thus, patients with MS should seek medical advice before receiving vaccines [12,28].

## Conclusion

We found that the prevalence of COVID-19 in the patients with MS was higher than in the other studies, which could be an alarm for patients with chronic and immunosuppressive diseases in the healthcare system in Iran.

In our study, men and employed people were more susceptible to COVID-19 infection, which could be due to their high exposure to SARS-CoV-2. The incidence of COVID-19 infection was much higher before vaccination, indicating an important role of vaccination in COVID-19 prevention. This study showed that a significant percentage of the patients with MS received at least one dose of vaccine, recalling the important role of vaccination in the control of COVID-19. The AstraZeneca vaccine caused the most common side effects among vaccines, so people should be aware of vaccine side effects before receiving vaccination. Future studies should address severe side effects to make more evidence-based decisions.

## Acknowledgments

The researchers would like to thank all the patients with multiple sclerosis for participating in this study. This study was approved by the neurology research center of Kerman University of Medical Sciences.

## Author Contributions

**Data curation:** Shamimeh Bahmani, Simin Jafari.

**Formal analysis:** Parya Jangipour Afshar.

**Methodology:** Simin Jafari.

**Project administration:** Behnaz Sedighi, Simin Jafari.

**Supervision:** Behnaz Sedighi, Aliakbar Haghdoost.

**Writing – original draft:** Shamimeh Bahmani, Simin Jafari.

**Writing – review & editing:** Behnaz Sedighi, Aliakbar Haghdoost, Parya Jangipour Afshar, Zohre Abna.

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
