## [Decision Letter · Decision Letter 0]

19 Sep 2022

PONE-D-22-22791Multiple sclerosis and covid-19: a retrospective study in IranPLOS ONE

Dear Dr. Jafari,

Thank you for submitting your manuscript to PLOS ONE. After careful consideration, we feel that it has merit but does not fully meet PLOS ONE’s publication criteria as it currently stands. Therefore, we invite you to submit a revised version of the manuscript that addresses the points raised during the review process.

We look forward to receiving your revised manuscript.

Kind regards,

Anza Bilal Memon, MD

Academic Editor

PLOS ONE

Journal Requirements:

Reviewers' comments:

Reviewer's Responses to Questions

**Comments to the Author**

1. Is the manuscript technically sound, and do the data support the conclusions?

Reviewer #1: Partly

Reviewer #2: Partly

2. Has the statistical analysis been performed appropriately and rigorously? 

Reviewer #1: I Don't Know

Reviewer #2: No

3. Have the authors made all data underlying the findings in their manuscript fully available?

Reviewer #1: No

Reviewer #2: No

4. Is the manuscript presented in an intelligible fashion and written in standard English?

Reviewer #1: No

Reviewer #2: No

5. Review Comments to the Author

Reviewer #1: First, the grammar is very poor. The entire manuscript needs a significant revision for grammar.

Regarding the content, there are several major problems.

1. What does COVID infection mean? Does it mean symptomatic + PCR confirmation? Does it mean antigen testing? Is it patient reported or physician confirmed?

2. Employed vs. unemployed needs to be further explored. At a condition that associated with COVID, were employed people working in a private office or in a crowded market with exposure to many other people? Were unemployed people confined to their homes or moving around their urban centers?

3. What is "addiction"? Addiction to what - smoking?

4. What is "physical disease"?

5. What medications were these patients using for their MS?

6. Other risk factors for COVID include diabetes, hypertension, lung disease, etc - did these patients have those?

Reviewer #2: Sedighi et al present a retrospective evaluation of CoVID-19 related outcomes in patients with MS in a referral clinic located in Kerman, Iran. The authors assessed factors associated with COVID-19 infection, along with the effectiveness and untoward effects of the CoVID-19 vaccine. It is helpful to get a report on the impact of CoVID19 on the Iranian MS population. There are several points that should be addressed in the article:

1. The references for the global prevalence of MS are not current and should be updated. The most recent data puts the world MS prevalence estimate at 2.8 million based on the 3rd edition of the Atlas of MS (Walton C, et al. Mult Scler. 2020 Dec; 26(14): 1816–1821. doi: 10.1177/1352458520970841).

2. How representative is the 367-study sample of the Kerman MS clinic population? How many patients were excluded either due to failure to obtain consent or for other reasons? Was this a convenience population based on lab testing or clinic follow-ups? These issues should be documented and explained.

3. How was SARS-CoV2 infection confirmed in the study population and over what period of time? It is difficult to judge the 35% incidence of “getting CoVID-19” in the study population if the details are not presented. Was the diagnosis made in laboratory-confirmed patients based on either polymerase chain reaction or serology tests vs. presumptive diagnosis based on symptoms or exposures? Also, the time period of tracking infections is important with the arrival of various CoVID-19 variants which have different infectivity rates. These data should be documented in the methods.

4. The MS disability status (EDSS) is obtained on only 57/367 study participants. This is missed opportunity to assess how neurological disability impacts the study outcomes. In several studies in other countries, neurological disability made CoVID19 infection more severe and more likely to be recognized. On a related note, I’m wondering if the MS population at the clinic was more severely impaired neurologically than comparable MS populations with such a high (69%) unemployment rate in a young adult population.

5. Were MS disease modifying therapies documented in the study population? If not, why not. Patients with MS taking CD20 disease modifying therapies (e.g., rituximab and ocrelizumab) have been shown to be at increased risk for CoVID-19 and have less robust responses to vaccines.

6. Significant digits after the decimal point for odds ratios and confidence intervals should be limited to 1 or 2 (e.g., 1.1 or 1.11).

7. Clear labeling of the risk factor of interest should be presented in the abstract in results as opposed to the generic variable label. Male sex vs. sex and current employment vs employment status.

6. PLOS authors have the option to publish the peer review history of their article (what does this mean?). If published, this will include your full peer review and any attached files.

Reviewer #1: No

Reviewer #2: No

---

## [Author Response · Author response to Decision Letter 0]

14 Nov 2022

Thank you for your note. The manuscript is based on the formatting guidelines.

We will upload data as Supporting Information files.

It was corrected in the manuscript.

Review Comments to the Author

Reviewer #1: 

First, the grammar is very poor. The entire manuscript needs a significant revision for grammar.

Thank you for your note. We revised the manuscript for English language and grammar.

Regarding the content, there are several major problems. 

1. What does COVID infection mean? Does it mean symptomatic + PCR confirmation? Does it mean antigen testing? Is it patient reported or physician confirmed?

The diagnosis was based on the physician's judgment, which used lab tests, including PCR and clinical signs and symptoms.

2. Employed vs. unemployed needs to be further explored. At a condition that associated with COVID, were employed people working in a private office or in a crowded market with exposure to many other people? Were unemployed people confined to their homes or moving around their urban centers?

Employed means having a job working for a company or another person with usual exposure to other people, while unemployed means not having a permanent job or being a university student and also may have usual exposure to other people. 

We didn’t feel the need for further exploration with our patients since they were strict about self-protection and self-isolation due to their underlying disease. Conversely, our patients were completely informed on the protocols for preventing the contraction of COVID infection since the beginning of the pandemic. 

3. What is "addiction"? Addiction to what - smoking?

In this study, we asked the patients about their lifestyle habits, such as tobacco, alcohol, and substance abuse (opioid, heroin, methamphetamine, or other stimulant substance). We added details in the methods section.

4. What is "physical disease"? 

In this study, we only recruited subjects without any important chronic diseases, such as diabetes, hypertension, cardiovascular, cerebrovascular, and respiratory diseases, and "physical disease" means complications accompanying MS other than the conditions noted above.

5. What medications were these patients using for their MS?

The patient's medications were added in table 1.

6. Other risk factors for COVID include diabetes, hypertension, lung disease, etc - did these patients have those?

The patients who had chronic non-communicable diseases (diabetes, hypertension, cardiovascular, cerebrovascular, and respiratory diseases) and those who did not agree to participate in the study were excluded.

Reviewer #2:

 Sedighi et al present a retrospective evaluation of CoVID-19 related outcomes in patients with MS in a referral clinic located in Kerman, Iran. The authors assessed factors associated with COVID-19 infection, along with the effectiveness and untoward effects of the CoVID-19 vaccine. It is helpful to get a report on the impact of CoVID19 on the Iranian MS population. There are several points that should be addressed in the article:

1. The references for the global prevalence of MS are not current and should be updated. The most recent data puts the world MS prevalence estimate at 2.8 million based on the 3rd edition of the Atlas of MS (Walton C, et al. Mult Scler. 2020 Dec; 26(14): 1816–1821. doi: 10.1177/1352458520970841).

Thank you for your note. We updated the MS prevalence.

2. How representative is the 367-study sample of the Kerman MS clinic population? How many patients were excluded either due to failure to obtain consent or for other reasons? Was this a convenience population based on lab testing or clinic follow-ups? These issues should be documented and explained.

We used the registry of MS patients who received care from the only referral hospital in Kerman province. Out of 548 registered cases, we recruited 367 (66.9 %); those who did not agree or had a major non-communicable disease such as diabetes, hypertension, cardiovascular disease, cerebrovascular disease, and respiratory disease were excluded. We added details in the methods section. 

3. How was SARS-CoV2 infection confirmed in the study population and over what period of time? It is difficult to judge the 35% incidence of “getting CoVID-19” in the study population if the details are not presented. Was the diagnosis made in laboratory-confirmed patients based on either polymerase chain reaction or serology tests vs. presumptive diagnosis based on symptoms or exposures? Also, the time period of tracking infections is important with the arrival of various CoVID-19 variants which have different infectivity rates. These data should be documented in the methods.

The patients with COVID-19 in this study were those who had the disease since the beginning of the pandemic that includes various CoVID-19 variants and the diagnosis was based on physician judgment, which used lab tests including PCR and clinical signs and symptoms. We added details in the methods section.

4. The MS disability status (EDSS) is obtained on only 57/367 study participants. This is missed opportunity to assess how neurological disability impacts the study outcomes. In several studies in other countries, neurological disability made COVID-19 infection more severe and more likely to be recognized. On a related note, I’m wondering if the MS population at the clinic was more severely impaired neurologically than comparable MS populations with such a high (69%) unemployment rate in a young adult population.

Based on the information in table 1, EDSS was obtained from all the patients, of whom 309 were categorized as mild and 57 as moderate to severe.

The EDSS provides a total score ranging from 0 to 10: 0-2.5 refers to people with a mild degree of disability, 3-5 refers to people with a moderate degree of disability, and 5.5-10 refers to severe disability.

We must add that the majority of the samples are women and unemployment is not just because of their MS conditions, a considerable proportion of healthy young adult women or university students do not have full-time jobs, so the unemployment rate in our sample is high.

5. Were MS disease modifying therapies documented in the study population? If not, why not. Patients with MS taking CD20 disease modifying therapies (e.g., rituximab and ocrelizumab) have been shown to be at increased risk for CoVID-19 and have less robust responses to vaccines.

Table. Type of DMT and getting COVID- 19 in each group

 Covid-19 

DMT Yes No P-value

Injectable 49 86 0.14

Anti CD20 23 40 

S1P receptor modulator (Fingolimod) 20 22 

Natalizumab 2 16 

Oral (Dimethyl fumarate, Teriflunomide) 23 48 

None 11 25 

In response to the comment, we assessed and recorded the history of patients including disease modifying therapies. In summary, and incompatible with our expectation, we did not find any significant difference in contracting covid-19 between users and non-users of these drugs (table above), however, some finding was also reported in some references in the following.

Based on the above explanation, we added the following sentences in the result section.

We compared the prevalence of COVID-19 in patients with MS based on the type of DMT, our result showed that there is not a significant difference in the prevalence of COVID-19 according to their medications.

1. Bsteh G, Assar H, Hegen H, Heschl B, Leutmezer F, Di Pauli F, Gradl C, Traxler G, Zulehner G, Rommer P, Wipfler P. AUT-MuSC investigators. COVID-19 severity and mortality in multiple sclerosis are not associated with immunotherapy: Insights from a nation-wide Austrian registry. PLoS One. 2021;16(7): e0255316.

2. Louapre C, Collongues N, Stankoff B, Giannesini C, Papeix C, Bensa C, Deschamps R, Créange A, Wahab A, Pelletier J, Heinzlef O. Clinical characteristics and outcomes in patients with coronavirus disease 2019 and multiple sclerosis. JAMA neurology. 2020 Sep 1;77(9):1079-88.

6. Significant digits after the decimal point for odds ratios and confidence intervals should be limited to 1 or 2 (e.g., 1.1 or 1.11).

 It was corrected in the manuscript.

7. Clear labeling of the risk factor of interest should be presented in the abstract in results as opposed to the generic variable label. Male sex vs. sex and current employment vs employment status.

 It was corrected in the abstract section.

---

## [Decision Letter · Decision Letter 1]

28 Dec 2022

PONE-D-22-22791R1Multiple sclerosis and covid-19: a retrospective study in IranPLOS ONE

Dear Dr. Jafari, 

Thank you for submitting your manuscript to PLOS ONE. After careful consideration, we feel that it has merit but does not fully meet PLOS ONE’s publication criteria as it currently stands. Therefore, we invite you to submit a revised version of the manuscript that addresses the points raised during the review process. The authors have made some significant changes to the manuscript, as recommended by the reviewers. However, I agree with reviewer # 2 that it requires some additional changes to refine the manuscript to meet the standards of PLOS One Journal publication. These changes are not difficult to make and should be done soon, and revised version should be submitted for my review and approval. Authors must pay special attention to manuscript formatting and grammar. I suggest that authors should seek some professional help for manuscript editing. Please carefully read the comments of reviewer # 2 and respond accordingly with your manuscript edits.

Please submit your revised manuscript by 1/18/2023 If you will need more time than this to complete your revisions, please reply to this message or contact the journal office at plosone@plos.org. Please include the following items when submitting your revised manuscript:A rebuttal letter that responds to each point raised by the academic editor and reviewer(s). You should upload this letter as a separate file labeled 'Response to Reviewers'.A marked-up copy of your manuscript that highlights changes made to the original version. You should upload this as a separate file labeled 'Revised Manuscript with Track Changes'.An unmarked version of your revised paper without tracked changes. You should upload this as a separate file labeled 'Manuscript'.

We look forward to receiving your revised manuscript.

Kind regards,

Anza Bilal Memon, MD

Academic Editor

PLOS ONE

Reviewers' comments:

Reviewer's Responses to Questions

**Comments to the Author**

1. If the authors have adequately addressed your comments raised in a previous round of review and you feel that this manuscript is now acceptable for publication, you may indicate that here to bypass the “Comments to the Author” section, enter your conflict of interest statement in the “Confidential to Editor” section, and submit your "Accept" recommendation.

Reviewer #1: (No Response)

Reviewer #2: (No Response)

2. Is the manuscript technically sound, and do the data support the conclusions?

Reviewer #1: (No Response)

Reviewer #2: Partly

3. Has the statistical analysis been performed appropriately and rigorously? 

Reviewer #1: (No Response)

Reviewer #2: No

4. Have the authors made all data underlying the findings in their manuscript fully available?

Reviewer #1: (No Response)

Reviewer #2: Yes

5. Is the manuscript presented in an intelligible fashion and written in standard English?

Reviewer #1: (No Response)

Reviewer #2: Yes

6. Review Comments to the Author

Reviewer #1: (No Response)

Reviewer #2: While many of the comments were addressed in the revision, there are still issues that remain in the updated manuscript. The authors should address the following:

1. Significant grammatical errors remain in the revised article making several sections difficult to read.

2. In the abstract, it would be more informative to include quantitative numerical descriptions rather than qualitative statements. For example, in the abstract it states that Kerman is “one of the largest cities in Iran.” It would be helpful if the abstract mentioned the numerical population of the city. Instead of stating “most of the patient with multiple sclerosis in this study received at least one dose of the vaccine,” it would be helpful to state what percentage of the patients in the study received the vaccine.

3. In the first paragraph of the results section, it would be helpful to list the mean EDSS score of the patient and explain how the patients’ disease activity was stratified into mild, moderate or severe based on the EDSS score.

4. The table in the results section references "physical disease,” but this term is not further explained in the results or the methods sections. Without further explanation of this term earlier in the paper, the reader will have difficulty understanding what this term refers to.

5. In the results section, it states that there was no difference in the prevalence of COVID-19 between the different DMT groups. It would be helpful to mention in the discussion section whether Anti CD20 agents were included. If anti-CD 20s were not included, it may be helpful to discuss that this may have impacted the data as other studies have demonstrated that Anti-CD use impacted vaccine efficacy.

6. Table 5 lists the number of adverse effects, but the types of adverse effects were never defined. It would be beneficial to list in the table the different types of adverse effects that were included.

7. PLOS authors have the option to publish the peer review history of their article (what does this mean?). If published, this will include your full peer review and any attached files.

Reviewer #1: **Yes: **Michael Levy

Reviewer #2: No

---

## [Author Response · Author response to Decision Letter 1]

20 Jan 2023

While many of the comments were addressed in the revision, there are still issues that remain in the updated manuscript. The authors should address the following:

1. Significant grammatical errors remain in the revised article making several sections difficult to read.

Thank you for your note. We revised the manuscript for English grammar.

2. In the abstract, it would be more informative to include quantitative numerical descriptions rather than qualitative statements. For example, in the abstract it states that Kerman is “one of the largest cities in Iran.” It would be helpful if the abstract mentioned the numerical population of the city. Instead of stating “most of the patient with multiple sclerosis in this study received at least one dose of the vaccine,” it would be helpful to state what percentage of the patients in the study received the vaccine.

It was revised in the abstract section.

3. In the first paragraph of the results section, it would be helpful to list the mean EDSS score of the patient and explain how the patients’ disease activity was stratified into mild, moderate or severe based on the EDSS score.

It was revised in the method and result section.

4. The table in the results section references "physical disease,” but this term is not further explained in the results or the methods sections. Without further explanation of this term earlier in the paper, the reader will have difficulty understanding what this term refers to.

 It was corrected in the manuscript.

5. In the results section, it states that there was no difference in the prevalence of COVID-19 between the different DMT groups. It would be helpful to mention in the discussion section whether Anti CD20 agents were included. If anti-CD 20s were not included, it may be helpful to discuss that this may have impacted the data as other studies have demonstrated that Anti-CD use impacted vaccine efficacy.

It was added in the discussion.

6. Table 5 lists the number of adverse effects, but the types of adverse effects were never defined. It would be beneficial to list in the table the different types of adverse effects that were included.

It was added in the table.

---

## [Editor Report · Decision Letter 2]

2 Mar 2023

PONE-D-22-22791R2Multiple sclerosis and covid-19: a retrospective study in IranPLOS ONE

Dear Dr. Jafari,

Thank you for submitting your manuscript to PLOS ONE. After careful consideration, we feel that it has merit but does not fully meet PLOS ONE’s publication criteria as it currently stands. Therefore, we invite you to submit a revised version of the manuscript that addresses the points raised during the review process.

We look forward to receiving your revised manuscript.

Kind regards,

Anza Bilal Memon, MD

Academic Editor

PLOS ONE

Journal Requirements:

Additional Editor Comments:

Results:

Line 134-136 please re-write the paragraph as listed below:

Furthermore, we compared the prevalence of COVID-19 in patients with MS based on the type of DMT, and our results showed no significant difference in the prevalence of COVID-19 based on the kind of DMT taken by the patients.

Keep R as upper case for reference throughout the tables. This needs to be done for all tables.

Discussion:

Line 177-179: Please add a little discussion here about the effects of CD 20 and COVID-19 severity in your patient population and how it affected the vaccine efficacy in your patient population?

Conclusion:

Line 197-199 Please re-write paragraph as listed below:

We found that the prevalence of COVID-19 in the patients with MS was higher than in the other studies, which could be an alarm for patients with chronic and immunosuppressive diseases in the healthcare system in Iran.

---

## [Author Response · Author response to Decision Letter 2]

9 Mar 2023

Journal Requirements:

Thank you for your note. The reference list has been checked and corrected.

Additional Editor Comments:

Results:

Line 134-136 please re-write the paragraph as listed below:

Furthermore, we compared the prevalence of COVID-19 in patients with MS based on the type of DMT, and our results showed no significant difference in the prevalence of COVID-19 based on the kind of DMT taken by the patients.

It was corrected in the manuscript.

Keep R as upper case for reference throughout the tables. This needs to be done for all tables.

Discussion:

Line 177-179: Please add a little discussion here about the effects of CD 20 and COVID-19 severity in your patient population and how it affected the vaccine efficacy in your patient population?

 It was added in the discussion.

Conclusion:

Line 197-199 Please re-write paragraph as listed below:

We found that the prevalence of COVID-19 in the patients with MS was higher than in the other studies, which could be an alarm for patients with chronic and immunosuppressive diseases in the healthcare system in Iran.

It was corrected in the manuscript.

---

## [Editor Report · Decision Letter 3]

12 Mar 2023

Multiple sclerosis and covid-19: a retrospective study in Iran

PONE-D-22-22791R3

Dear Dr., Jafari, 

We’re pleased to inform you that your manuscript has been judged scientifically suitable for publication and will be formally accepted for publication once it meets all outstanding technical requirements.

Within one week, you’ll receive an e-mail detailing the required amendments. When these have been addressed, you’ll receive a formal acceptance letter, and your manuscript will be scheduled for publication.

An invoice for payment will follow shortly after the formal acceptance. To ensure an efficient process, please log into Editorial Manager at http://www.editorialmanager.com/pone/, click the 'Update My Information' link at the top of the page, and double check that your user information is up to date. If you have any billing related questions, please contact our Author Billing department directly at authorbilling@plos.org.

Kind regards,

Anza Bilal Memon, MD

Academic Editor

PLOS ONE
---

## [Editor Report · Acceptance letter]

16 Mar 2023

PONE-D-22-22791R3 

Multiple sclerosis and COVID-19: a retrospective study in Iran 

Dear Dr. Jafari:

I'm pleased to inform you that your manuscript has been deemed suitable for publication in PLOS ONE. Congratulations! Your manuscript is now with our production department. 

Kind regards, 

on behalf of

Dr. Anza Bilal Memon 

Academic Editor

PLOS ONE